# Reconstruction and Removal Mechanisms of Gel-like Membrane Fouling for Seawater Desalination: Experiments and Molecular Dynamics Simulations

**DOI:** 10.3390/polym14183734

**Published:** 2022-09-07

**Authors:** Qi Wang, Xiangyu Yang, Ronghui Qi, Lizhi Zhang

**Affiliations:** 1School of Electric Power Engineering, South China University of Technology, Guangzhou 510640, China; 2Key Laboratory of Enhanced Heat Transfer and Energy Conservation of Education Ministry, School of Chemistry and Chemical Engineering, South China University of Technology, Guangzhou 510640, China

**Keywords:** seawater desalination, artificial contaminant gel, molecular dynamics simulation, effects of detergent

## Abstract

Anti-gel fouling is a key problem faced by membrane desalination, especially for applications in organic acid-rich seawater. In this paper, a chemical crosslinking-based method was used to reconstruct and characterize the gel pollutants produced under the actual operating conditions of seawater desalination. In addition to the calcium alginate/calcium humate three-dimensional network skeleton, salt ions (K^+^, Na^+^, Mg^2+^, Cl^−^) in solution were also considered to ensure that the reconstructed gel was similar to pollutant gels on membranes under practical operating conditions. Characterizations showed that the reconstructed gel has high thermal insulation and stability, thus adjusting the temperature has no removal effect. Two detergents (sodium citrate and sodium hydroxide) were investigated, and their gel-removal mechanism was elucidated by molecular dynamics simulation. Numerical analysis showed that the electrostatic attraction interaction had a significant role in the gel cleaning process. Owing to the attraction of the lower electrostatic potential region in the cleaning agent, the ion exchange between Na+ in the cleaning agent and Ca^2+^ in the gel led to the breaking of the Ca^2+^-induced intermolecular bridge in the complex. As the adhesion of fouling gels decreased, the gel water solubility was increased, resulting in a decrease in weight and strength of the gel. Therefore, the integrity of the gel fouling layer was weakened and can be effectively removed. This study provides a theoretical basis for the removal of gel-like membrane fouling during actual seawater desalination.

## 1. Introduction

As an effective means to solve water shortage, many desalination technologies have been developed, including multi-stage flash distillation and multi-effect distillation (MED), reverse osmosis, nanofiltration and electrodialysis, and membrane distillation [1]. Among them, the membrane-based humidification/dehumidification seawater desalination (MHDD) system has emerged as a new water treatment technology that combines thermal and membrane methods. MHDD operates on the principle that by using a heat source to increase the water vapor partial pressure on the seawater side, the resulting vapor pressure difference on both sides allows water vapor to permeate through the porous hydrophobic membrane, whereas substances such as salt in seawater cannot. Subsequently, the vapor can be condensed to obtain high-purity fresh water. The advantages of the MHDD system include the ease of using low-grade energy sources, such as solar energy, because of its operating temperature (between 70 °C and 90 °C). Owing to the mild vaporization process used, the scale of the system equipment is relatively small [2,3,4,5]. In addition, its operating pressure is close to atmospheric pressure.

However, during the long-term operation of the system, the membrane surface will be contaminated by wetting or the mutual attraction between the membrane and impurities, so the pollution resistance of the membrane needs to be considered. Membrane pollution can generally be divided into three categories: inorganic pollution, organic pollution, and biological pollution [6]. The seawater environment is not a pure water solution, but a mixture of various ions. Organic pollutants can easily stick to the membrane surface, rapidly increasing the polluted area. The existence of organic pollution will accelerate the formation of mixed pollutants, which is difficult to remove by simple mechanical means. Choudhury et al. [7] examined the interactions between contaminants and different types of MD membranes and the influence of different operating conditions on the occurrence of fouling and wetting. Membrane wetting will let the direct permeation of the feed solution through the membrane pores, resulting in reduced contaminant rejection and overall process failure. Jia et al. [8] attached a thin super-hydrophilic coating to the superhydrophobic film to construct a film with an asymmetric structure, which reduced pollution and scaling and exhibited higher and more stable vapor flux than the original PTFE film. Qin et al. [9] examined the synergistic effect of combined fouling in the MD process with three organic foulants in the presence of colloidal silica particles. Results showed that the combined organic fouling with colloidal silica particles not only deteriorated water production but also compromised product quality by partial membrane wetting. Jia et al. [10] provided a robust rGO composite membrane with high flux and anti-wetting properties in membrane distillation and offered some insight into the anti-wetting mechanism of rGO against SDS and water. Shao et al. [11] carried out appropriate charge modification of neutral ultrafiltration membranes to better reduce membrane pollution caused by electrostatic interaction and the comprehensive effect of membrane aperture.

In general, contamination by salt and small amounts of gel can be largely circumvented using hydrophobic membranes. However, the eutrophication of seawater has become a major water pollution issue in many countries [12]. Eutrophic seawater is rich in alginic acid and humic acid, which can combine with Ca^2+^ and other divalent metal ions to form gels in large quantities according to their specific structure [13,14]. Gel contamination by eutrophic seawater is particularly challenging to overcome. When seawater desalination is inevitable, and the nearby seawater is severely eutrophic, it is particularly important to study the performance of the pollutant gel layer and remove it. Even during the normal real-world seawater desalination process, organic pollution is dominant [15]. The main frame of the gel is usually a calcium alginate or calcium humate three-dimensional network structure, and the network is filled with solution [16]. The gel in organic pollution is the most difficult pollutant to remove due to its special structure. The gel not only has strong viscoelastic adhesion but also continues to increase, eventually leading to the blockage of membrane pores, covering the membrane surface and greatly reducing the permeability flux. For contaminant gels, it is desirable to consider the addition of inhibitors during formation or the use of appropriate cleaning methods to clean the gel and restore membrane properties. The addition of the inhibitor prevents gel formation and fundamentally solves the problem of gel contamination. Membrane cleaning is mainly to restore the initial flux of the membrane as much as possible by destroying the solute layer adsorbed on the membrane surface or removing impurities in the membrane pores.

Generally, there are three cleaning methods of cleaning gel contaminants from membranes: salt washing, osmotic reflux, and chemical cleaning [17]. In general, the cleaning efficiency of cleaning methods follows the order: osmosis backwash > chemical cleaning > salt cleaning. Chemical cleaning is a more suitable method for cleaning organic dirt, but the chemical cleaning process destroys the active layer of the membrane, thus reducing its durability. Salt cleaning and osmotic backwash remove Ca^2+^-induced bridged dirt more effectively than chemical cleaning. Salt washing is more convenient and efficient because the operation does not require high-pressure devices. Lee et al. [18] investigated the cleaning of organic-fouled reverse osmosis membranes with concentrated salt solutions. Daly et al. [19] systematically examined how organic fouling characteristics and osmotic backwashing parameters influence cleaning efficiency. Tu et al. [20] simulated four typical membrane fouling conditions under controlled laboratory conditions. Zazouli et al. [15] found that individual alginate fouling was more detrimental than individual humic acid fouling. Hao et al. [21] evaluated the effect of organic substances on the fouling behavior of a thin film composite (TFC) membrane with in situ Ca^2+^ addition.

As gel is a type of soft matter, it is a special disperse system. The macromolecular organic acid particles in the solution are interconnected by divalent metal ions such as Ca^2+^ to form a three-dimensional network structure in the whole system, and the liquid is wrapped in it. When the gel is formed, the system not only loses its fluidity but also exhibits the mechanical properties of a solid, such as elasticity and strength. Gel is a colloidal dispersion system; its properties fall between those of a solid and a liquid. Therefore, the gel is quite different from most other inorganic pollution and biological pollution. Based on the above reasons, the action mechanism of the cleaning agent on the gel fouling layer is still unclear. An advantage of molecular simulation is that it can study the structural changes of the gel layer more deeply, at the microscopic and molecular levels, in the process of heat and mass transfer [22,23,24,25]. Stewart et al. [26] studied the binding of sodium and calcium ions to single and multiple poly-G decamer strands by conducting a series of molecular dynamics simulations. The results revealed the binding modes that provide a rationale for the observed gelling of alginate by calcium rather than sodium ions. Hecht et al. [27] used molecular dynamics simulation to investigate the binding of sodium and calcium ions to alginate chains, as well as its dependence on the content of guluronic (G) acid residues. The results revealed the association behavior that has a clear dependence on G content and the nature of interactions of sodium and calcium ions are shown to differ for poly-M, poly-G, and the heteropolymer compositions. Xiang et al. [28] used a steered molecular dynamics (SMD) approach to investigate the molecular binding mechanism between the polyamide (PA) membrane and the alginate molecules in the presence of Ca^2+^ ions in an aqueous solution. Stewart et al. [29] studied the effect of the alginate amphiphilic structure on membrane fouling by molecular dynamics. Quantum chemical calculations on the M and G monomers of alginate reveal that M adopts an equilibrium geometry that is hydrophilic on one face and hydrophobic on the other, which is potentially amphiphilic.

Previous studies on the pollutant gel layer produced by seawater desalination were mainly experimental studies. Most of the experimental studies use seawater or homemade sea water and have a long experiment cycle. It is necessary to prepare an artificial gel with the same physical and chemical properties as real-world pollutant gels. This study used a chemical cross-linking method to reconstruct the gel and tested its physical and chemical properties by a variety of characterization methods. The effect of two optimal cleaning agents on gel structure was analyzed by molecular dynamics simulation (MDS). We revealed the mechanism of colloid disintegration on the membrane surface, and the results will provide guidance for membrane cleaning or membrane modification.

## 2. Experiments and Methods

### 2.1. Materials

Sodium alginate (≥99%, M_w_ = (198.11)_n_, SA, reagent grade), humic acid (FA ≥ 90%, HA), calcium chloride (≥96%, 110.98, CaCl_2_), potassium chloride (≥99.5%, 74.55, KCl), sodium chloride (≥99.5%, 58.44, NaCl), magnesium chloride hexahydrate (≥98%, 203.3, MgCl_2_·6 H_2_O), sodium hydroxide (≥97%, 40, NaOH), magnesium sulfate (≥99.5%, 120.37, MgSO_4_), hydrochloric acid (0.1 mol/L, HCl), calcium bicarbonate (≥99%, 162.06, Ca(HCO_3_)_2_), sodium bicarbonate (≥99.5%, 84.01, NaHCO_3_), sodium citrate tribasic dihydrate (294.1, AR), deionized water (DW) were obtained from Guangzhou Qianhui Co., Guangzhou, China.

### 2.2. Synthetic Gel Fabrication Processes

In line with the composition and proportion of membrane gel fouling formed by the seawater of Lantau Island, Hong Kong, and Yantian, Shenzhen measured by air sweep membrane distillation (SGMD), after comprehensive measurement and proportion adjustment, the synthetic gel was prepared by a chemical cross-linking method. The components and proportions of gel pollutants produced by the two different seawaters are shown in Table 1.

First, 5 g sodium alginate (NaAlg) and 5 g humic acid (HA) solid powder were added into a beaker with 200 g deionized water. The components were dispersed for 2 h using a KQ-100VDV type three-frequency CNC ultrasonic cleaner, then stirred for 1 h by an hJJ-6A multi-head digital display constant-temperature magnetic stirrer. After the powder was completely dissolved, 0.6 g KCl, 19.2 g NaCl, and 2.4 g MgCl_2_ were successively added according to the measured seawater pollutant sample composition ratio. The HJJ-6A multi-head digital display constant-temperature magnetic agitator was used to stir the mixture until the solid particles were completely dissolved. Next, the mixed solution was left to debubble for more than 48 h. The solution was applied evenly on the glass plate with a glass rod and placed in anhydrous CaCl_2_ solution with a mass fraction of 10%, prepared in advance, and left for (48–72) h for static crosslinking. Finally, the cross-linked gel was gently removed from the glass plate, and the resulting finished pollutant gel was stored in anhydrous CaCl_2_ solution with a mass fraction of 1%. The whole process was performed at 25 °C. Pure alginate and pure humic acid gels were prepared by the same method.

Physical photo of the gel is shown in Figure 1. The gel prepared in this paper can be regulated according to the composition and proportion of pollutant gel produced by different water sources, but the preparation time was much less than the formation time of gel in real-world situations. Moreover, it can be made in large quantities, thereby avoiding the traditional method of breaking down components to obtain real gels.

### 2.3. Characterization of the Synthetic Gel

The contact angle of the gel layer was measured by a JC2000Cl contact angle system (Shanghai Zhongchen Digital Technology Equipment Co., LTD., Shanghai, China) with a 3 μL water droplet. Each sample was measured three times and the average contact angle was used. The surface topology of the gel layer was characterized by atomic force microscopy (AFM, Multimode 8, Bruker, Billerica, MA, USA). The scan size was 5 μm × 5 μm. Scanning electron microscopy (SEM, Merlin, Zeiss, Oberkochen, Germany) was used to observe the gel layer morphologies. A poly crystal X-ray diffraction (XRD, X’pert Powder, PANalytical, Almero, The Netherlands) was performed to analyze the gel layer crystalline microstructure. The 2θ angular range was selected between 10° and 90°. Infrared spectroscopies of the gel layer were obtained in KBr pellets using a Nicolet IS50-Nicolet Continuum FTIR spectrometer (Thermo Fisher Scientific, Waltham, MA, USA). The porosity and the pore size distribution were measured using a mercury injection apparatus (AutoPore IV 9500, Micromeritics Instrument Corp, Atlanta, GA, USA). The thermal conductivities of the gel layer were measured by the thin film module of the Hot Disk Thermal Constants Analyzer (hot disk, TPS 2500S, Hot Disk LTD., Uppsala, Sweden). The stress–strain, elastic modulus, and fracture energy of the gel were tested by electronic universal testing machine (WDW-200, Jinan East Testing Instrument Co., LTD., Jinan, China). The thermal stability and phase transition of the gel were tested by a TG-DSC synchronous thermal analyzer (STA449F3, NETZSCH-Gerätebau GmbH, Selb, Germany).

### 2.4. Gel Disintegration Experiment

Typically, the reason for the sharp drop in water production is that the contaminant gel on the membrane is a dense polymer. Inhibition experiments were performed to identify inhibitors before gel formation. The inhibition experiment involved the addition of a certain concentration of salt ions to the cross-linking agent before gel formation. Dense degree is defined as the mass of the substance formed after the inhibitor was added divided by the mass of the gel. Owing to the requirement for a high level of adding inhibitors and the great influence of the seawater environment, such as osmotic pressure and concentration, cleaning disintegration tests were performed on the formed gel instead. In the cleaning experiment, acid and alkali solutions often mentioned in the literature were used as the basic comparison items, and different salt solutions were used to treat the gel. The disintegration degree is defined as the dissolved mass divided by the total mass of gel. Sodium citrate as a cleaning agent was rarely seen in the literature; citric acid solution was usually more common.

## 3. Molecular Dynamics Simulations

### 3.1. Mathematical Model

The total potential energy of a system can be calculated from the sum of valence (or bond), cross term, and non-bond interactions:(1)$$E_{\mathrm{total}}{=E}_{\mathrm{valence}\;}{+E}_{\mathrm{cross}}{+E}_{\mathrm{non}-\mathrm{bond}}$$
(2)$$E_{\mathrm{valence}}{=E}_{\mathrm{bond}}{+E}_{\mathrm{angle}}{+E}_{\mathrm{torsion}}{+E}_{\mathrm{oop}}$$
(3)$$E_{\mathrm{non}-\mathrm{bond}}{=E}_{\mathrm{vdW}}{+E}_{\mathrm{Coulomb}}{+E}_{h-\mathrm{bond}}$$

The valence energy, *E_valence_* (KJ/mol), is generally accounted by four terms: bond stretching term (*E_bond_* (KJ/mol)), valence angle bending term (*E_angle_* (KJ/mol)), dihedral bond-torsion term (*E_torsion_* (KJ/mol)), out-of-plane interaction term (*E_oop_* (KJ/mol)). *E_cross_* (KJ/mol) represents cross-term interacting energy which reflects coupling motions between the adjacent bonds in a molecule. Finally, *E_non-bond_* (KJ/mol) is the non-bond interaction term which includes the van der Waals energy (*E_vdW_* (KJ/mol)), the Coulomb electrostatic energy (*E_Coulomb_* (KJ/mol)), and the hydrogen bond energy (*E_h-bond_* (KJ/mol)). The more detailed mathematical expressions of the potential for the COMPASS force field had been cited by Chang et al. [30].

The radial distribution function *g*(*r*) defines the probability density of finding a pair of particles with centers at a distance of *r* from each other. It takes a molecule or atom in the system as the target unit, the distance between the molecule and the center is d*N* from *r* → *r* + d*r*, then *g*(*r*) can be expressed as [31]:(4)$$g(r)=\frac{dN}{{4\pi r}^{2}\rho dr}$$

Its value can be understood as the ratio of regional density to average density, that is, the probability of occurrence. The higher the *g*(*r*), the higher the probability.

According to Hildebrand’s solubility formula, the relationship between interaction parameters and solubility parameters between polymer and solvent can be obtained [32]:(5)$$\chi =\frac{V_{R}}{\mathrm{RT}}{{(\delta }_{A\;}-{\;\delta }_{B})}^{2}$$

*χ* is the interaction parameter between polymer and solvent. *δ* ((KJ/cm^3^)^1/2) is the solubility parameter, which is defined as the square root of cohesion density:(6)$$\delta =\sqrt{\mathrm{CED}}=\sqrt{\frac{\Delta E}{\tilde{V}}}$$
where *CED* (KJ/cm^3^) is cohesive energy density, that is, evaporation energy of solvent per unit volume, Δ*E* (KJ) is cohesive energy, and $\tilde{V}$ (cm^3^) is the volume of the amorphous cubic element.

According to the normal solution theory [33]:(7)$$\Delta {H=\Phi }_{1}\Phi_{2}{{(\delta }_{1}-\delta_{2})}^{2}$$
(8)ΔG=ΔH−TΔS
where, ΔH (KJ/mol) is the mixing enthalpy, $\Phi_{1}{\;\mathrm{and}\;\Phi }_{2}$ are volume fraction of mixture, ΔG (KJ/mol) and ΔS (KJ/mol·K) are mixing free energy and mixing entropy, respectively.

### 3.2. Polymer Structure Reconstruction

In this study, the time step was 1.0 fs. In order to avoid surface effects, periodic boundary conditions were prescribed in all directions to simulate the properties. All molecular dynamic simulations were performed using the Materials Studio 2019 software package of Accelrys, Inc. (San Diego, CA, USA). The COMPASS (Condensed-phase Optimized Molecular Potentials for Atomistic Simulation Studies) force field [34,35] was applied for all simulations. In the simulation procedure, pressure and temperature were controlled by the Berendsen barostat [36] and the Andersen thermostat [37], respectively. The COMPASS force field is a powerful force field to optimize the structural, conformational, vibrational, and thermophysical properties of a broad range of molecules, including polymer and organic–inorganic systems.

In the literature, alginate polymer chains consisting only of α-L-Gulonuronic acid (G) monomer were mainly studied because the G monomer was involved in gel formation. The main components of the gel in this paper consist of linear alginate polymer chains (containing β-D-mannuronic acid (M) and α-L-Gulonuronic acid (G) monomers) and humic acid (a complex mixture, a representative TNB structure was selected).

We considered G segments, M segments, and the crosslinking between the two chains in alginate polymer at the same time. Each alginate polymer chain has 96 repeating units and a total of 4 polymer chains formed a network structure. The reason is that there is random ordering between the two monomers, and different spacing or different spatial positions of carboxylic acid functional groups bound with cations will affect the final result [27]. Then, according to the composition and the proportion of each ion, added humic acid, KCl, NaCl, MgCl_2,_ CaCl_2,_ and water molecules. It was close to the actual structure of the gel. One of the monomer models of alginate polymer is shown in Figure 2a. TNB structure of humic acid is shown in Figure 2b. The molecular model of small molecules is shown in Figure 2c–f.

First, molecules mentioned above were randomly packed in an initial simulation unit with a size of 5.1 × 5.1 × 4.5 nm^3^. The unit had 11,712 atoms overall. It was under periodic boundary conditions at a temperature of 333.15 K. The initial structure was unstable due to the unreasonable structural distribution of the molecular chains in the initial state. To eliminate the unstable conformations, after geometrical optimizations by the smart minimizer method, the initial polymer structure was annealed from 333.15 K to 500 K over a period of 1000 ps in NVE (N: constant particle number, V: constant volume, E: constant energy) ensemble and 500 ps in an NPT (P: constant pressure) ensemble. The cell dimensions changed as the molecular structures became equilibrated. The final structure was obtained when the density of the cell agreed well with the experimental value of the gel. Finally, the x, y, and z dimensions of the polymer model approached 50.43 Å, 50.43 Å, and 44.26 Å, respectively. This was the final size of the calculation domain.

### 3.3. Simulation Procedures

The data model was used to model the thermal conductivity of the gel. Here, the RNEMD was adopted to calculate the thermal conductivity via Fourier’s law [38,39]. The method worked by exchanging velocities between two molecules in different parts of the simulation cell. In this paper, the simulation box had periodic boundary conditions in the three directions of the space, which was divided into N (N = 40) slabs with the same thickness of 1.26 Å along the heat flux (z) direction. The first and the last slabs of the box were designated to be the hot slabs while the N/2 slab was designated to be the cold slab. To maintain a heat flux, the fastest particle in the cold slab was replaced by the slowest particle in the hot slab and vice versa. This swapping was repeated periodically every 600 time steps. The main reason for choosing this swapping (W = 600) was that it ensures a rapid convergence of the simulated temperature gradient and produces a linear response within the simulation cell. Another reason was to ensure the average temperature of the simulation box was close to the ambient temperature of the experiment. As the swapping continued, the hot region became colder, whereas the cold region increased in temperature; a temperature gradient in the system was established due to the energy flux from the hot slabs to the cold ones. When the change in energy flux was less than 5% of the value before the change, the heat transfer had reached an equilibrium state and a steady temperature profile was obtained. The average energy flux in the steady state can be written as:(9)$$J_{z}=\frac{1}{2\mathrm{tA}}\underset{\mathrm{transfers}}{\sum }\frac{m}{2}{(v}_{\mathrm{hot}}^{2}-{\;v}_{\mathrm{cold}}^{2})$$
where *t* (s) is the total simulation time; *A* (m^2^) is the cross-sectional area of the simulation box perpendicular to the flow direction *z*; *m* (g/mol) is the atomic mass; $v_{hot}^{2}$ is the velocities of the coldest particle (smallest kinetic energy) in the hot slab. $v_{cold}^{2}$ is the velocities of the hottest particle (largest kinetic energy) in the cold slab. Transfers represent that the energy was transferred from the hot slab to the cold slab.

The thermal conductivity of the gel layer is calculated by RNEMD as shown in Equations (9)–(11). After achieving a steady state, the temperature *T* (K) in each slab was calculated by:(10)$$T=\frac{1}{{3nk}_{B}}\overset{n}{\underset{i=1}{\sum }}m_{i}v_{i}^{2}$$
where the sum extends over the atoms *i* in slab *k* with masses $m_{i}$ and velocities $v_{i}$; n is the number of atoms in the slab; $k_{B}$ is the Boltzmann constant, (1.38 × 10^−23^ J/K). According to the obtained temperature field above, the thermal conductivity λ (W/m·K) of the gel layer can be calculated from Fourier’s law, given as:(11)$$\lambda =-\frac{J}{\partial T/\partial x}$$

The radial distribution function (*RDF*) between oxygen atoms (central ions) and other ions (Ca^2+^ or Na^+^) was used to analyze the ordered structures generated by the central ions. The ordered structure showed clear peak characteristics corresponding to the characteristic distance between the particles and the central atoms. RDF was modeled and calculated by adding different contents of Na^+^ or K^+^ into the simulation unit, followed by the addition of chloride ions to neutralize electricity. Finally, the dynamics calculation was performed, and the RDF was analyzed by the Forcite module of MS.

The CED of the optimized model was calculated by the Forcite module of MS, from which the solubility parameters *χ* were obtained. Common parameters (elastic moduli, Poisson’s ratio, volume moduli, and shear moduli) of the gel were calculated by the mechanical properties module of MS [40].

Electrostatic interaction is an important long-range interaction between molecules. The electrostatic potential distribution of molecular systems affects the conformational orientation and chemical reaction activity, which can be used to analyze the interaction sites and mechanisms between clusters. Therefore, the electrostatic potential of a molecular system is one of the important parameters in describing the physical and chemical properties of molecules. In practical systems, the electrostatic potential of the molecular van der Waals surface and beyond is of great chemical significance. In the molecular field, atomic point charges are usually used to fit and reproduce the surface electrostatic potential distribution of molecular systems.

## 4. Experimental Results

To comprehensively consider the properties of the complex, real-world pollutant gels, and eliminate the uncertainty caused by different dosage, the research object of this paper was the gel formed by alginate and humic acid at a concentration ratio of 1:1, as shown in Figure 1.

### 4.1. Characterization Results of Reconstructed Gel

To confirm the physical properties of the prepared gels were consistent with the contaminant gels of real seawater desalination process and can replace the real-world gel for subsequent analyses [41], wide-angle X-ray diffraction (XRD) was performed to study the synthesis of gel crystalline microstructure, as shown in Figure 3a. The two peaks at 26.5° and 44° are characteristic peaks of calcium alginate [42]; 26.5° corresponds to the characteristic eggshell structure of calcium alginate, indicating that the pollutant gel layer has the characteristics of calcium alginate. Similarly, the characteristic peaks of humic acid were observed near 16.5° and 25°, and the characteristic bands of sodium chloride were found at 18.5°, 31°, 52°, and 71.5°. The characteristic peak band near 28.5° was potassium chloride. Characteristic peaks of magnesium chloride appeared near 21° and 33°. Through comparison, it can be found that the main characteristic peaks of the artificial gel prepared in this study are close to those of corresponding pollutants produced by DCMD experiments applied to real seawater in the previous study [41]. Through the analysis of the characteristic peaks of the above substances, it can be seen that there was no obvious overlap or annexation of each peak zone. This indicates that there was no significant chemical reaction between the substances free from the gel skeleton and the frame substance and that they maintain their own properties, which should have an impact on the physical, chemical, and permeability properties of the gel. According to the unique water absorption and retention properties of the gel, the more substances trapped inside the gel, the higher the osmotic pressure, and the more likely the gel will swell and increase its thickness. The thickening of the gel fouling layer is obviously detrimental to the moisture permeability of the permeable membrane.

In the infrared spectrum, the wide and strong absorption band at 3433 cm^−1^ represents the -OH stretching vibration peak on the calcium alginate molecular chain, as shown in Figure 3b. The stretching vibration of -OH groups in the six-membered ring of alginate macromolecule was limited by the formation of the “egg-box” structure by Ca^2+^ complexation. The wider width of the vibration peak indicates that only some hydroxyl groups participate in coordination, and other hydroxyl groups associate with each other, forming absorption at the wave number and overlapping with the low wave number, resulting in a wider peak shape. The corresponding absorption peak at 1030 cm^−1^ is the stretching vibration peak of C-O in the gel. Owing to the formation of -C-O-Ca-O-C-O- group structure by calcium crosslinking with alginate or humic acid, the stretching vibration absorption of C-O is strong [43]. The strong absorption at 1629 cm^−1^ represents the asymmetric stretching vibration of -COO^−^. The strong absorption at 1409 cm^−1^ represents the symmetric stretching vibration of -COO^−^ [44]. The presence of hydroxyl and carboxyl groups endows gel pollutants with strong hydrophilic properties. This is consistent with the contact angle results of gels described below. Hydrophilicity undoubtedly exacerbates the pollution of conventional hydrophobic membranes (such as PVDF membranes), which can only be avoided by hydrophobic modification. Based on the above analysis of the characteristic peaks of gel element composition, it can be seen that the substance in the solvent and calcium alginate skeleton should be combined mainly through hydrogen bonding and van der Waals force, without obvious chemical reaction.

To analyze the effect of the gel layer on the whole system, the surface structure of the gels was observed by scanning electron microscopy (Appendix A), and the pore structure of the gel layer was measured by the mercury injection method. The porosity of the gel layer is shown in Table 2 and the pore size distribution is shown in Figure 4. As can be seen from Figure 4, the pore sizes of the gels of different samples are all distributed around (20–150) nm. As can be seen from Table 2, Figure 4 and Appendix A, the gel layer has small porosity and a very small average pore size. Compared with PVDF membranes commonly used in seawater desalination (average pore size is approximately 511.5 nm and porosity is approximately 77.76%), which allows high steam permeability, the existence of a gel layer will undoubtedly increase pollution and lead to a sharp decrease in permeability.

Then, as shown in Appendix A, the three-dimensional rough structure of the pollutant gel layer was observed through AFM imaging. The scan size was 5 μm × 5 μm. The mean roughness (Ra) obtained from the AFM analysis was used to indicate the roughness of the pollutant gel layer surfaces; this is listed in Table 2. It can be observed that the surface roughness of the gel layer was low by the three-dimensional morphology. If a relatively flat gel layer is firmly attached to the membrane, it will not only prevent the infiltration of water vapor but is also difficult to remove by simple mechanical washing.

The relationship between the contact angle of membranes and wettability is given by Young’s equation [45]. The surface is hydrophilic when a contact angle is less than 90°. The surface is hydrophobic when a contact angle is greater than 90°. Surface contact angle (SCA) was measured to elucidate the hydrophilic and hydrophobic properties of the synthesized gel. Table 2 lists these values. Generally, the gel layer was hydrophilic. This can explain that when the gel is initially attached to the membrane surface, the membrane is extremely wettable, leading to increased contamination. Although the gel as a whole presents hydrophilic characteristics, according to the literature, the humic acid gel also contains some hydrophobic groups, which confers amphiphilic characteristics and makes it difficult to remove [46]. The hydrophilicity and hydrophobicity of a gel can provide basic parameters for the regulation of the gel contamination layer in subsequent periods.

### 4.2. Physical Properties of Reconstructed Gel

According to the above analysis of the operating principle of the MHDD system for seawater desalination, when the thermal conductivity of gel contamination is low, increased heat loss occurs, which is not conducive to the water treatment process and results in wasted resources. Through the tests by the Hot Disk Thermal Constants Analyzer, the effective thermal conductivity of the gel layer was approximately (0.02876–1.1538) Wm^−1^K^−1^. The total thermal resistance r (kW^−1^m^2^K) of the gel can be calculated from Equation (12). This calculation shows the total heat resistance can reach (0.17334–6.954) KW^−1^m^2^K. Usually, if the thermal conductivity of the material is not greater than 0.12 Wm^−1^K^−1^, the material is considered a thermal insulator, and if the thermal conductivity of the material is 0.05 Wm^−1^K^−1^ or below, it is considered an efficient thermal insulation material. Although there was a wide range of thermal conductivity values for the gels, most of the measured gels were below 0.12 Wm^−1^K^−1^ (over 70%); thus, the heat insulation performance of the gel layer was relatively high. Usually, the PVDF membranes used in the MHDD system have a thermal conductivity of approximately 0.05523 Wm^−1^K^−1^. Through the comparison of the above parameters, the mass transfer resistance was not only caused by the low porosity, but also the decrease of the temperature difference on both sides of the membrane caused by the low heat conduction of the gel layer.
(12)$$r=\frac{\delta }{\lambda }$$

To understand the mass change, decomposition behavior, and thermal feedback of the gel layer under different temperature water treatments, a TG-DSC synchronous thermal analyzer was used to analyze the gel. Thermal stability is a very important performance index for gel clearance. If the gel has low thermal stability, its structure will be destroyed at high temperature and its integrity cannot be maintained. Therefore, gel contamination on the membrane surface can be removed by appropriate heating. The thermogravimetric results of the gel are shown in Figure 5, from which the cleavage of the gel could be divided into four stages. The first stage occurs at (0–100) °C, the gel presents as endothermic and loses the water of crystallization near 100 °C. At this stage, the weight loss rate of gel increased, and was approximately 55%. In the second stage, (100–200) °C, the glycosidic bond in the gel framework breaks, decarboxylation occurs, and the adjacent hydroxyl groups were removed in the form of water molecules, with a weight loss rate of approximately 15%. In the third stage, (200–550) °C, the carboxyl group in the gel cracked at high temperature, releasing CO_2_, and the product was partially carbonized with further weight loss of approximately 8%. In the fourth stage, (550–900) °C, the carbides in the gel continue to crack, mainly CaCO_3_ cracking and eventually forming CaO, with further weight loss of approximately 3%. Between 100 °C and 200 °C, the gel lost the water of crystallization, and the structure began to change. The weight loss rate in the third and fourth stages was slow (the weight loss rate of the gel was 11%, which only accounts for 24% of the non-water compounds (which accounts for 45% of the gel)), indicating the sufficient thermal stability of the gels. However, MHDD systems typically operate at temperatures between 40 °C and 70 °C; gels in this temperature range are thermally stable enough to withstand the damage caused by high temperatures. Therefore, once the gel was generated in its original environment, it was in a stable state and gradually accumulates. It is difficult to remove or reduce gel contamination through heating at this temperature range.

To propose follow-up measures to remove or avoid the stubborn gel layer, the stress and strain characteristics of the gel were tested, as shown in Figure 6. As shown in Figure 6a,b, when the gel aggregation reached a certain degree, the stress and fracture energy of the gel layer increased as the concentration of the alginate increased. The strength and adhesion of the gel pollution in the later stage will increase. The elastic modulus of the gel and the variation in the trends of stress in the gel over temperature were analyzed, as shown in Figure 6c,d. As can be seen in Figure 6c, the energy storage modulus and loss modulus tended to be stable under a certain shear stress level; a sudden drop occurred only at a certain extreme stress level. This indicated that the gel layer is a viscoelastic material. The coexistence of viscosity and elasticity result in its strong tensile strength and compressive stress resistance. It also shows that the gel layer has good mechanical properties which can maintain its shape to a certain extent and the gel layer is not easily deformed by external influences. As shown in Figure 6d, the stress of the gel was still around 10 Pa, although it decreased within the temperature range of (40–70) °C, which is commonly used for seawater desalination. Combined with the analysis of the above parameters, simple physical and mechanical cleaning cannot completely remove the gel, so it is necessary to avoid gel formation from the source or to find a stronger, more effective cleaning method. In addition, the use of ordinary membrane materials cannot avoid surface adhesion.

### 4.3. Disintegration Experiments before and after Gel Formation

The results of adding inhibitors before gel formation are shown in Figure 6a. When a certain concentration of salt ions (K^+^ or Na^+^) was added to the solvent water, a small amount of loose flocculation was formed instead of ordinary, lumped gels, corresponding to the reduction in dense degree shown in Figure 7a. As shown in the figure, the density of floccule formed after the addition of salt inhibitors was greatly reduced, which has an obvious effect on gel prevention. However, in the actual system operation, adding inhibitors will change the osmotic pressure of seawater, and the pressure in the equipment for processing needs to be increased, which is not worth the loss. The experimental results of cleaning after gel formation are shown in Figure 7b below. Among them, the selected cleaning agent (C) were: 1, magnesium sulfate; 2, hydrochloric acid; 3, calcium bicarbonate; 4, sodium bicarbonate; 5, sodium hydroxide; 6, sodium citrate. As shown in Figure 7b, the chemical stability of gel in C5 and C6 solution was poor; that is, the disintegration degree was high. C5 was an alkali solution. When an alkali was applied to the gel, it catalyzes the oxidation of the gel, resulting in splitting of the macromolecular chain of the gel and in the disintegration of the gel. The cleaning effect was not as good as that of sodium citrate (C6). During the experiment, it was observed that when the temperature of the treatment solution gradually increased, the solution also gradually turned yellow, confirming the existence of an oxidation reaction, but also showed that with the increase in temperature and the extension of treatment time, this oxidation was more intense. C6 is a salt solution, sodium citrate binds to Ca^2+^ in the solution to form a soluble complex. The driving force for the complexation of calcium ions in the generation of thermodynamically more stable Ca-complex ions in the solution. Because the electrochemical polarization of the ionic Ca complexes was increased, the complexed form was more stable [47]. After citrate has combined with metal ions, the resulting complexes cannot be dissociated easily, reducing the free Ca^2+^ that can form a gel, and achieving the highest cleaning intensity. These two different types of cleaning fluids can be selected for molecular dynamics simulation. In addition, the experimental observation showed that as the salt solution concentration, temperature, or treatment time increased, the disintegration rate of the gel structure would accelerate.

## 5. Simulation Results

### 5.1. Verification of Model

The established model was compared with experiments to verify the correctness of the model. The gel density in the experiment was approximately 1.80 g/cm^3^, and the simulated density calculated by the molecular model was 1.81 g/cm^3^, as shown in Figure 8a, which was in good agreement with the experimental data. The thermal conductivity of the gel ranged from 0.028 to 1.154 Wm^−1^K^−1^ measured in the experiments. When the heat flux converged to 5.38 × 10^8^ W/m^2^ (Figure 8b), the effective thermal conductivity of the gel calculated by MDS using the RNEMD method was 0.25 Wm^−1^K^−1^, as shown in Figure 8c. This was within the range of experimental measurements. This confirmed that the thermal conductivity values were in agreement with the experimental data.

### 5.2. Results and Discussion

Because of the better effect of the two types of cleaning agents containing Na ions in the experiments, Na ions were used for calculation in the simulation. To compare the two different types of cleaning agents, we calculated NaOH and sodium citrate based on the experimental results. As shown in Figure 9, the radial distribution function of relevant elements before and after the gel was cleaned was calculated. According to the RDF results, the Na^+^ in sodium citrate or NaOH was closer to the O atom on the organic acid skeleton. Na atoms in the cleaning agents replaced Ca atoms and were closer to the O atoms that would have formed the gel, resulting in a loose, disintegrating gel. This meant that the gel dense network structure was destroyed after cleaning with sodium citrate or NaOH. According to the r value of curves in Figure 9a,b, it can be inferred that although the alkali solution permitted Na atoms closer access to the central oxygen atom that would have formed the gel due to the catalytic action, the concentration of free Ca^2+^ does not continuously decrease but was in a state of dynamic equilibrium, so its height g(r) was much smaller than that of the sodium citrate (After citrate has combined with metal ions, the resulting complexes cannot be dissociated easily, reducing the free Ca^2+^ that can form a gel, and achieving the highest cleaning intensity). Based on the above analysis, it can be concluded that sodium citrate had a stronger effect than NaOH in the process of gel disintegration. This is consistent with the experimental results in Section 4.3 above.

The RDF parameters were then used to analyze the distance between Ca^2+^ and oxygen atoms in organic acid molecules at different sodium citrate concentrations. As shown in Figure 9c, the Ca^2+^ was further away from the gel-forming O atom in the organic acid chains as the concentration of the cleaning agent increased (1:1 → 2:1). The initial distribution of the r value increased from approximately 2.5 to 3.5. The g (r) at the top of the curve drops to approximately half of what it was before, indicating that Ca^2+^ was less likely to be present near the central O atoms. The value of g (r) increased the distribution to the right, as with high concentrations of the cleaning agent; that is, Ca^2+^ was dispersed farther away from the organic acid owing to the presence of a higher concentration of the cleaning agent. This further indicated that in addition to the type of cleaning agent, the concentration of cleaning agent also had a great influence on the change in gel crosslinking degree and the time of gel disintegration. The simulation results further verified the damage by cleaning agent to the gel structure in the gel disintegration experiment in Section 4.3 above. This provided ideas for follow-up studies on whether there was a synergistic effect that aggravated gel contamination in the presence of other ions and how various cleaning agents act on gel contamination layers.

The solubility parameter is an important parameter that indicates the strength of intermolecular interaction, which plays an important role in predicting the solubility of polymers in solvents. It is generally believed, in hybrid system, that when the solubility parameter difference between the polymer and solvent is |Δδ| < 4(J/cm^3^)^1/2^, the polymer and solvent are mutually soluble. When the difference of solubility parameters between the two is |Δδ| ≥ 10(J/cm^3^)^1/2^, it indicates that the polymer has solvent resistance. According to Equations (5) and (6), the solubility was simulated by CED in the Forcite module of MS, and the calculation results showed that the solubility parameter of water was 40, the solubility parameter of gel was 16, and the solubility parameter of sodium citrate was 19. As for the definition of solubility, the higher the *χ* value, the stronger the intermolecular force, and the lower the solubility. The calculation results show that the solubility parameters between the gel and the cleaning agent are very similar and the difference between them was only 12.5% of the difference between gel and water. According to the “like dissolves like” rule, the gel and cleaning agent can be mutually soluble.

To fully explain that the gel can disintegrate when contacted by the cleaning agent, the Gibbs free energy was calculated by density functional theory. In a certain thermodynamic process, the reduced internal energy of the system can be converted into part of the external work. The calculation results for Gibbs free energy are shown in Table 3. It can be seen that ΔG < 0; thus, the reaction happens spontaneously: the gel will dissolve spontaneously after contacting the cleaning agent. In the process of dissolution, Δ*G* < 0, TΔ*S* > 0, so Δ*H* < TΔ*S*. According to Formulas (5)–(8), Δ*H* > 0; therefore, if you want to have Δ*G* < 0, Δ*H* must be as small as possible. That is, the closer $\delta_{1}$ and $\delta_{2}$ are to each other, the more soluble they become.

The structural changes in the whole system during the cleaning process are shown in Figure 10. Figure 10a shows the initial stage of the system model construction, in which the gel and the cleaning agent were in an independent state. At this point, the monomer and chain of the gel form a tight cluster structure. For a clearly diagram, the chemical bonds of calcium elements in the gel and sodium elements in the cleaning agent were hidden. After the simulation was run for a period of time, the gel and cleaning agent gradually came into contact and the tight structure of the gel stretches, as shown in Figure 10b. In addition, Ca^2+^ were observed to move away from the gel structure and nearer to the cleaning agent sodium citrate. This process means that the cleaning agent gradually dissolved and broke down the dense structure of the gel.

The mechanical properties changes of this process were calculated by the mechanical properties module of Forcite in MS; the results are shown in Table 4. The elastic stiffness matrix of the gel in the calculation of mechanical properties is shown in Appendix A. Because the gels can dissolve completely or mostly in several of the cleaning agents mentioned, the degree of cleaning simulation was controlled by adjusting the concentration of the cleaning agent to approximately 50%. After the cleaning simulation, it was found that the bulk elastic modulus, shear elastic modulus, and Young’s modulus all decreased after cleaning, as shown in Table 4. In particular, the bulk modulus of elasticity has been reduced by less than half and the compression coefficient increased to more than twice the initial value. These results confirmed that cleaning had a large influence on gel disintegration.

According to the above experimental and related simulation parameters results, sodium citrate had the best cleaning and disintegrating effects on the gel. The solution after cleaning was complete without impurities and the distribution of the RDF constant was ideal. The molecular model of sodium citrate and two organic acids were geometrically optimized using the DMol3 module of the MS program to obtain their surface electrostatic potential. The electrostatic potential was to use a unit of “positive charge” to detect the electrostatic interaction potential around the molecule, which can directly reflect the characteristics of the charge distribution of the molecule itself. Electrostatic potential diagrams can be used to predict a variety of possible chemical properties of molecules, such as the effective site of action. The electrostatic potential distribution of sodium citrate and two organic acids were shown in Figure 11. The potential was not uniformly distributed but was locally distributed in several areas of high electrostatic potential (the red areas in the figure) and several regions of low electrostatic potential (the dark blue areas in the figure). Moreover, the cleaning agent (sodium citrate) had stronger negative charge sites than the two organic acids, that is, sodium citrate has a stronger attraction effect on Ca^2+^ than organic acids. Among them, the low electrostatic potential regions in sodium citrate were concentrated near the carboxylic acid groups and easily attract the calcium ions that were trapped in the gel. When the calcium in the gel was attracted to the sodium citrate, it formed a more stable soluble complex, which caused the gel’s network structure to gradually collapse, and the gel could not re-form owing to the gradual reduction in free Ca^2+^ in the solution. The analysis and calculation of electrostatic potential confirmed that sodium citrate more strongly adsorbs Ca^2+^ than organic acid and that the cleaning/disintegrating effect of sodium citrate on the gel was confirmed. The molecular mechanism of the interaction between the cleaning agent and gel was also explored through the calculation of this part. In the future research and development of new cleaning agents, we can therefore predict their effects through the calculations of electrostatic potential.

## 6. Conclusions

This paper provides a time-saving preparation method of gel-like membrane fouling for organic acid-rich seawater desalination by chemical cross-linking. In addition to calcium alginate/humic acid calcium skeleton, the presence of ions (K^+^, Na^+^, Mg^2+^, Cl^−^) was considered in the gel structure. Through characterization, this self-made pollutant was found to reproduce the characteristics of gel pollution under real operating conditions. The removal mechanisms were investigated by MDS and analysis. 

Results showed that the gel contaminants are viscoelastic (elastic modulus of 14.3713 GPa) and have high thermal insulation (thermal conductivity of 0.25 Wm^−1^K^−1^) and stability (solubility of 16), which cannot be removed by adjusting temperature. The coexistence of viscosity and elasticity leads to its strong tensile strength and compressive stress resistance. Two detergents (sodium citrate and sodium hydroxide) were investigated numerically. The solubility was close in the gel and the cleaning agent, and the difference between them was only 12.5% of the difference between gel and water. After contact with the cleaning agent, the bulk elastic modulus, shear elastic modulus, and Young’s modulus of the gel all decreased. The gel can dissolve spontaneously when contacting the cleaning agent due to the calculated Gibbs free energy of the contact process being ΔG < 0. Owing to the attraction of the lower electrostatic potential region in the cleaning agent, the ion exchange between Na^+^ in the cleaning agent and Ca^2+^ in the gel led to the breaking of the Ca^2+^-induced intermolecular bridge, the degree of polymerization of macromolecules in the gel was reduced. The adhesion of the fouling gel was weakened, and the water solubility of the gel was increased, resulting in a decrease in the weight and strength of the gel. Thus, the integrity of the gel fouling layer was weakened. These results are of great significance for future research into preventing gel formation or disintegrating gels after formation.

## Figures and Tables

**Figure 1 polymers-14-03734-f001:**
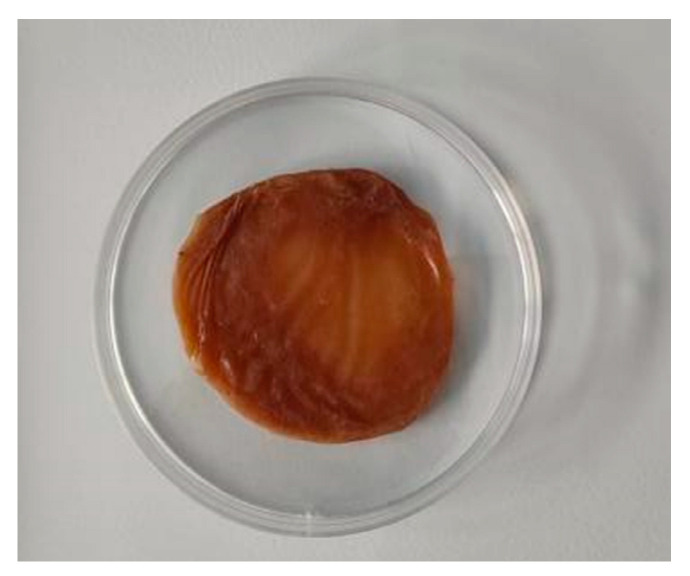
Physical photos of the gel (alginic acid to humic acid ratio was 1:1).

**Figure 2 polymers-14-03734-f002:**
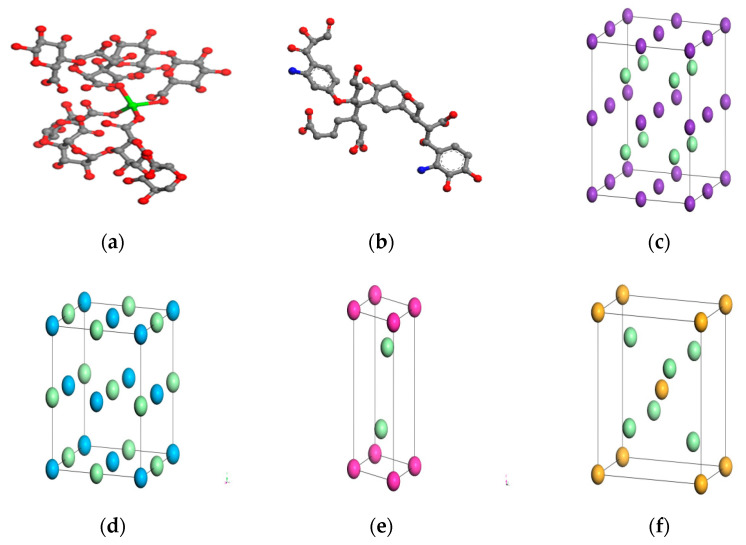
Molecular models of (**a**) monomer of an alginate polymer chain; (**b**) TNB structure of humic acid; (**c**) KCl; (**d**) NaCl; (**e**) MgCl_2_ and (**f**) CaCl_2_.

**Figure 3 polymers-14-03734-f003:**
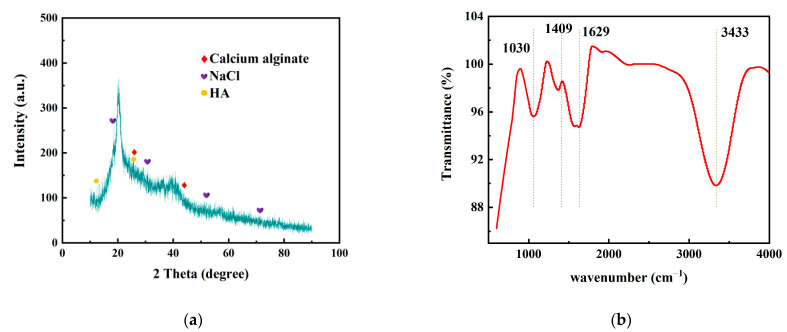
(**a**) The XRD analysis of gel. (**b**) The FTIR patterns of the gel.

**Figure 4 polymers-14-03734-f004:**
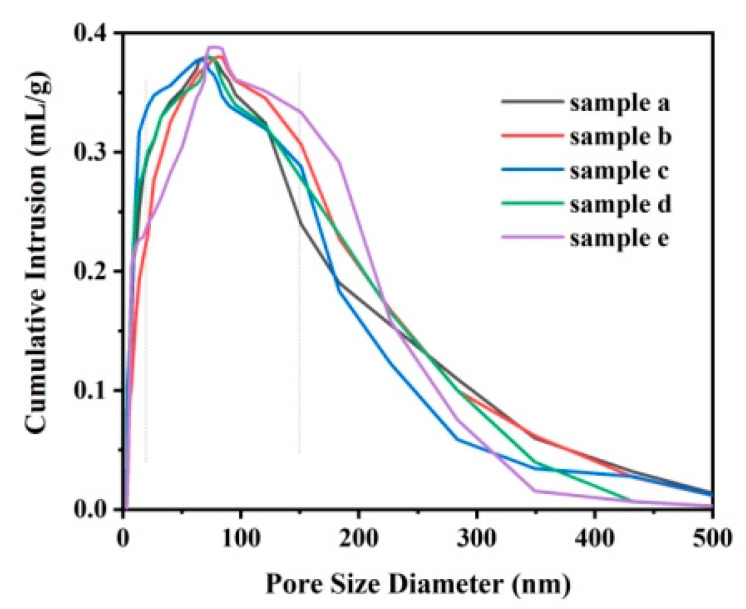
Distribution of pore size diameter of the gel.

**Figure 5 polymers-14-03734-f005:**
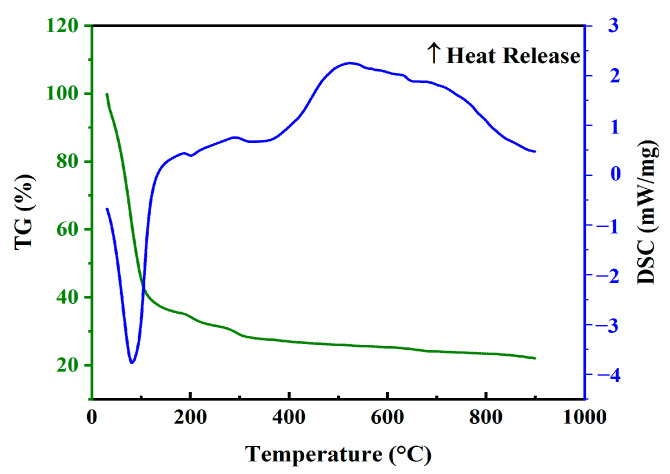
The thermal reactivity of the gel layer.

**Figure 6 polymers-14-03734-f006:**
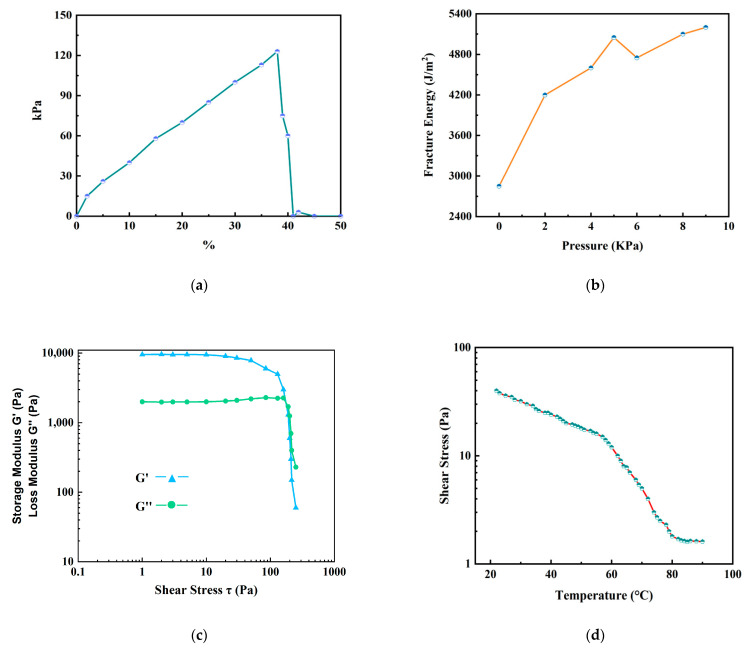
The stress–strain characteristic curve of gel layer: (**a**) the stress and strain of the gel; (**b**) the breaking energy of the gel; (**c**) the elastic modulus of the gel; (**d**) the tendency of the stress in the gel changing with temperature.

**Figure 7 polymers-14-03734-f007:**
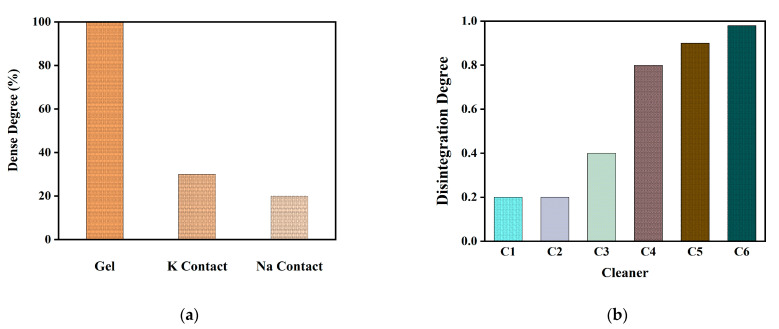
(**a**) The change of gel volume after adding certain concentration of salt ions. (**b**) Experimental results of pollutant gel cleaning.

**Figure 8 polymers-14-03734-f008:**
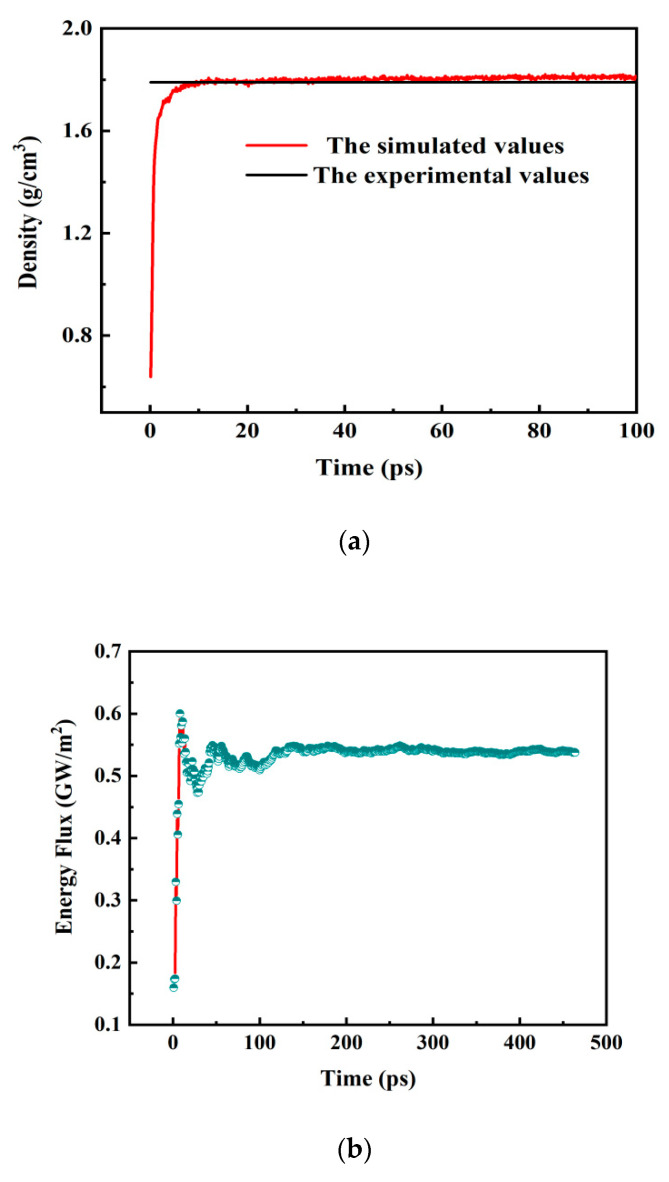
(**a**) The simulated density of gel. (**b**) The heat flux of the gel calculated by MDS; (**c**) the thermal conductivity of the gel calculated by MDS.

**Figure 9 polymers-14-03734-f009:**
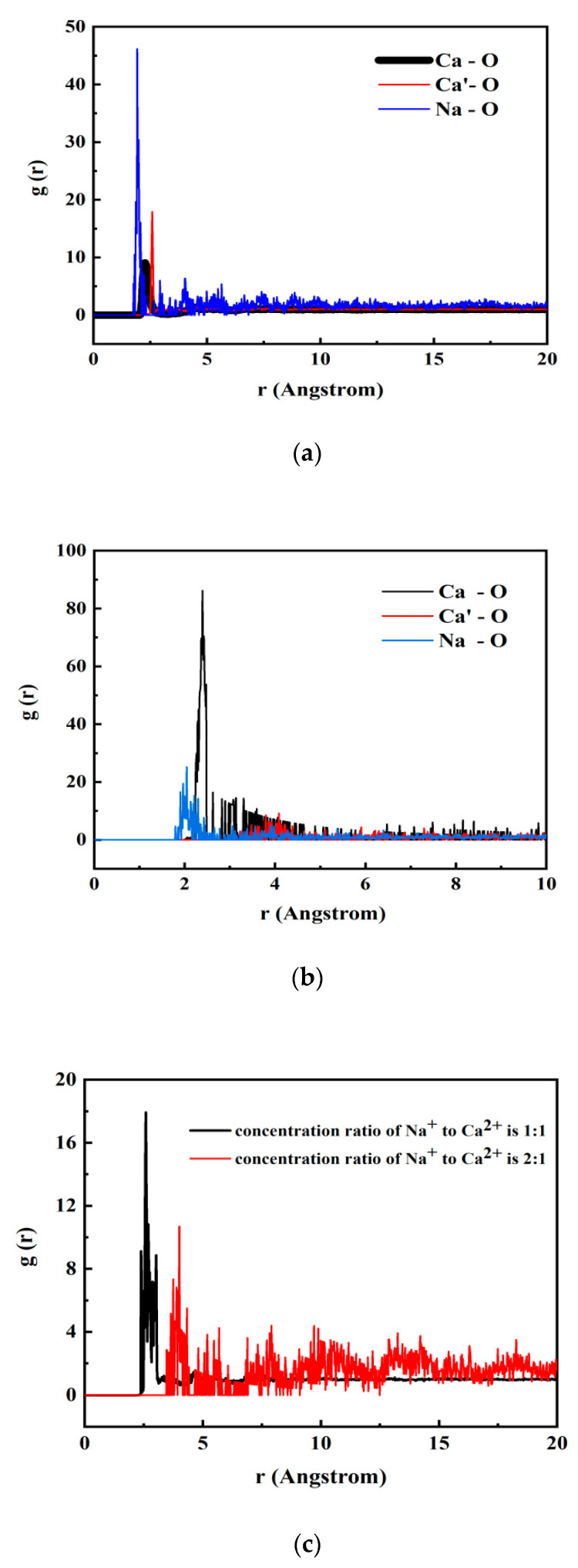
The RDF parameters before and after cleaning of (**a**) sodium citrate and (**b**) NaOH; (**c**) the RDF parameters at different sodium citrate concentrations.

**Figure 10 polymers-14-03734-f010:**
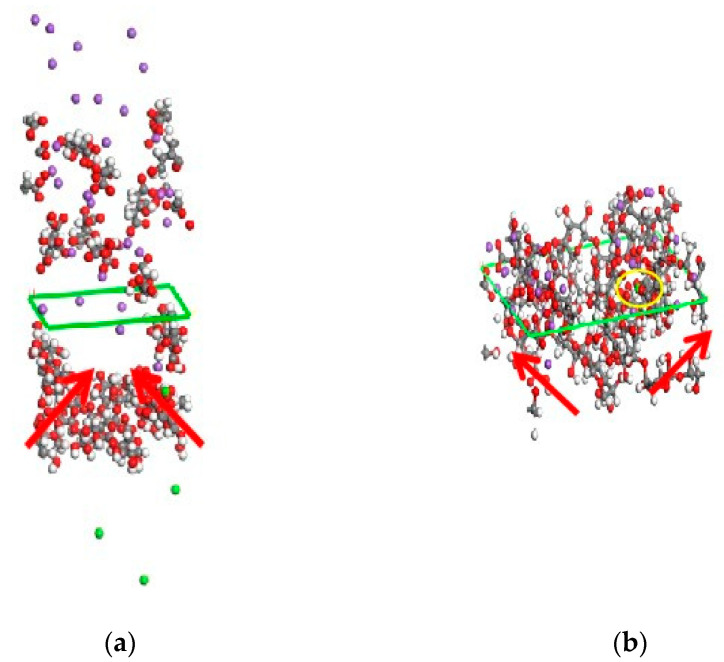
(**a**) The state of gel before contacting with cleaning agent; (**b**) the state of gel after contacting with cleaning agent.

**Figure 11 polymers-14-03734-f011:**
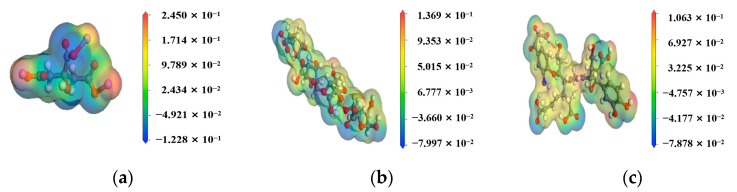
The electrostatic potential of (**a**) sodium citrate; (**b**) alginate acid; (**c**) humic acid.

**Table 1 polymers-14-03734-t001:** Composition of membrane gel fouling produced by two different seawaters.

Pollutants	Ion Concentration (mg/L)	
	Lantau Island	Yantian
K^+^	52.23	79.56
Ca^2+^	79.17	85.67
Na^+^	1011	2350
Mg^2+^	37.18	245.32
Cl^−^	2719	4620
Humic acid	890	550
Alginate	47.3	62.8

**Table 2 polymers-14-03734-t002:** The porosity, surface roughness (scan size 5 μm × 5 μm), surface contact angle of gel layer.

Different Samples	a	b	c	d	e
**Porosity (%)**	7	9.1	7.5	8.3	7.8
**Ra (nm)**	210	352	230	13.5	141
**Contact Angle (°)**	38.3	50	38.8	46.9	14.9

**Table 3 polymers-14-03734-t003:** The results of Gibbs free energy.

	Reactants		Products	
	3 Ca (Alg)_2_	2 Na_3_C_6_H_5_O_7_	6 NaAlg	Ca_3_ (C_6_H_5_O_7_)_2_
*E_total_* (Hartree)	−4069.714	−1313.146	−1856.116	−3698.666
*E_total_* (kcal/mol)	−2,553,783.978	−824,011.823	−1,164,730.179	−2,320,947.822
*G_total_* (298.15 K) (kcal/mol)	362.214	48.418	167.788	116.321
*E_τcorr_* (298.15 K) = *E_tota_*_l_ + *G_total_* (298.15 K) (kcal/mol)	−2,553,421.764	−823,963.4046	−1,164,562.391	−2,320,831.501
Δ*G_reaction_*	−13.744			

**Table 4 polymers-14-03734-t004:** Results of mechanical properties of gel and gel after cleaning (↓ indicates that the value decreased compared with that before cleaning. ↑ indicates an increase in the value compared with the value before cleaning).

	Gel	NaOH Cleaning Degree of 50%	NaHCO_3_ Cleaning Degree of 50%	Sodium Citrate Cleaning Degree of 50%
Bulk modulus of elasticity (GPa)	18.8398	8.9421↓	9.0861↓	8.8922↓
Elastic modulus of shear (GPa)	9.6313	7.2004↓	6.4532↓	6.4571↓
The compression coefficient (1/TPa)	53.333	113.3370↑	114.3476↑	114.0281↑
Young’s modulus (GPa)	22.0962	16.4021↓	15.6562↓	13.8669↓

## Data Availability

The data that support the findings of this study are available from the corresponding author, upon reasonable request.

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
