# Peer review of "Reconstruction and Removal Mechanisms of Gel-like Membrane Fouling for Seawater Desalination: Experiments and Molecular Dynamics Simulations"

_polymers, 2022, doi:10.3390/polym14183734_

Round 1

Reviewer 1 Report

The work studied the reconstruction and the removal mechanism of gel-like membrane fouling in seawater desalination by experimentally prepared gels and by simulations. Pollutant gels in membrane desalination are of great practical interests and is an active topics in the engineering of polymers. The presentation is detailed and clear, and the analysis is scientific sound. I consider the work is suitable for publication after the following concerns are addressed.

Figure 3a. The characteristic peaks claimed in the text for HA, Calcium alginate, NaCl (expect 18.5°) are merely visible from the plots. And it does not look similar to Ref [42] (presumably Fig 9b) where the amorphous peak was identified as part of the membrane itself, which is irrelevant here. Please clarify with these results.

In Fig 7a, the simulated values is slightly above the experimental values, while the text reports the former is 1.79 g/cm3 < 1.80 g/cm3, please check for probably misplaced data.

In Fig 8b, the Ca-O peaks clearly show artificial 1/r^2 height which is a sign of inadequate sampling and average. Please generate PDF with more independent samples for Fig 8b and probably also the read line in Fig S3 and check if the conclusion is still the same.

Also, Fig S3 is referred and analyzed in text. Better put this figure in the main text. 

Minor points.

L236, because: incomplete sentence/logic.

L295, the hot region become(s) colder.

L297, less than 5% (of what)

Author Response

Reviewer 1:

Question 1:

Figure 3a. The characteristic peaks claimed in the text for HA, Calcium alginate, NaCl (expect 18.5°) are merely visible from the plots. And it does not look similar to Ref [42] (presumably Fig 9b) where the amorphous peak was identified as part of the membrane itself, which is irrelevant here. Please clarify with these results.

Response:

Reference [42] refers to "Common parameters (elastic moduli, Poisson's ratio, volume moduli and shear moduli) of the gel were calculated by the mechanical properties module of MS [42]", and the "the amorphous peak was identified as part of the membrane itself" you refer to is Reference [43] 9 (B) XRD patterns of the membranes, which is irrelevant here. Figure 3 of this paper was compared with Figure 13 of reference [43]. Through comparison, it can be found that the the main characteristic peaks of the artificial gel prepared in this study are close to those of corresponding pollutants produced by DCMD experiments applied to real seawater in the previous study.

Question 2:

In Fig 7a, the simulated values is slightly above the experimental values, while the text reports the former is 1.79 g/cm3 < 1.80 g/cm3, please check for probably misplaced data.

Response:

We feel great thanks for your professional review work on our article. We recalculated the density according to the comments of the reviewers, and the simulated value was close to but slightly higher than the experimental value. For the error in the text report, we have corrected 1.79 g/cm3 in the text reports to 1.81g/cm3.

Question 3:

In Fig 8b, the Ca-O peaks clearly show artificial 1/r^2 height which is a sign of inadequate sampling and average. Please generate RDF with more independent samples for Fig 8b and probably also the read line in Fig S3 and check if the conclusion is still the same.

Response:

As you mentioned, the g (r) value around r=2.4 in Fig 8b before modification was not sufficiently sampled, so we regenerated the RDF curve of Ca-O with more independent samples as you suggested. As shown in the following figure, the Ca-O curve fluctuates around r=2.4 compared with the previous one, but the relative positions of the trends of the Ca-O, Ca'-O and Na-O curves are consistent with the previous results. Similarly, for Fig. S3, we performed similar sample addition collection. Compared with the original figure, the curve fluctuates near r=2.6, that is, near the highest value of g (r), but the relative positions of the two curves are close to the previous figure.

Figure 8 (b). The radial distribution function before and after cleaning of NaOH;

Figure 8(c). The RDF parameters at different sodium citrate concentrations (which is the previous figure S3.)

Question 4:

Also, Fig S3 is referred and analyzed in text. Better put this figure in the main text.

Response:

We are grateful for the suggestion. We have put Fig S3 into Figure 8c in the main text according to your comment.

Question 5:

Minor points.

L236, because: incomplete sentence/logic.

L295, the hot region become(s) colder.

L297, less than 5% (of what).

Response:

We have made corresponding modifications for the logic and language problems:

L233: The irrelevant statement "and because" was deleted.

L292: the hot region became colder.

L294: When the change in energy flux was less than 5% of the value before the change.

Reviewer 2 Report

The authors presented a study on the use of gel-like membranes for seawater desalination experiments. The authors also included MD simulations to study the effect of two optimal cleaning agents on the gel structure. The paper might be publishable after some suggestions/comments:

1) In Fig. 1., the authors might revise the label since the word organic acid is redundant.

2) In Fig. 3b, the wavenumber should be cut to 4000 cm^-1 - 500 cm^-1 and should be from 4000 to 500 not 0 - 5000.

3) The authors presented the pore size of the gel layer by SEM (Fig S1). They presented the porosity based on the mercury injection test but did not mention it in the methodology section. They should present the adsorption-desorption isotherms. 

4) In Table 2, the porosity percentage is not consistent with the increase of concentration of the organic samples. We could not deduce the relationship between the two. The same goes for the contact angle.

Author Response

Reviewer 2:

Question 1:

In Fig. 1., the authors might revise the label since the word organic acid is redundant.

Response:

Thanks for your correction and we have deleted the word organic acids and modified the label to: Physical photos of the gel (alginic acid to humic acid ratio was 1:1).

Question 2:

In Fig. 3b, the wavenumber should be cut to 4000 cm^-1 - 500 cm^-1 and should be from 4000 to 500 not 0 - 5000.

Response:

We have modified the figure. The modified figure is as follows:

Figure 3. (b) The FTIR patterns of the gel.

Question 3:

The authors presented the pore size of the gel layer by SEM (Fig S1). They presented the porosity based on the mercury injection test but did not mention it in the methodology section. They should present the adsorption-desorption isotherms.

Response:

We have already added mercury injection test in the methodology section: The porosity and the pore size distribution of the gels were measured using a mercury injection apparatus (AutoPore IV 9500). The adsorption desorption isotherm you mentioned is the gas adsorption method to measure the pore structure, and the mercury injection method is used in this structure. The pore size distribution of mercury injection method is shown in the following figure 4. Since the porosity measured by mercury injection method are independent data, they are listed in the table 2. The gel layer has small porosity (7%-9.1%) and the pore sizes of gels of different samples are all distributed around (20-150) nm. The gel layer will undoubtedly lead to a sharp decrease in permeability.

Figure 4. Distribution of pore size diameter of the gel.

Table 2. The porosity, surface roughness (scan size 5 μm × 5 μm), surface contact angle of gel layer.

Different samples

a

b

c

d

e

Porosity%

7

9.1

7.5

8.3

7.8

Ra(nm)

210

352

230

13.5

141

Contact angle (°)

38.3

50

38.8

46.9

14.9

Question 4:

In Table 2, the porosity percentage is not consistent with the increase of concentration of the organic samples. We could not deduce the relationship between the two. The same goes for the contact angle.

Response:

The values of porosity and contact angle are used to illustrate from multiple samples that the porosity of the gel is low (about (7 – 9.1) %), that the gel is hydrophilic (contact angle:(14.9 – 50) °) and indeed does not tend to change with increasing concentration. We have adjusted the different concentrations mentioned in this part to different samples (a-e).

Reviewer 3 Report

The current manuscript simulated the gel pollute commonly formed near/on membrane surfaces in water purification, and studied the effectiveness of chemical cleaning for removing this pollute. Although the topic of this work is very interesting for membrane-based water purification, there exists vague and messy parts in the current version. Therefore, I do not suggest it be published in “Polymers”.

Some comments are listed:

1.       What are the differences between salt washing and chemical cleaning?  The most chemical cleaning agents used in RO or NF are NaOH or acid, which are also considered salt.

2.       Membrane distillation is commonly used for saline feed, which is too harsh for RO, with potential higher pollutant concentration. However, in the current work, the authors used simulated “seawater” which is the feed for RO, not MD, as feed. Can the authors explain the reason behind it?

3.       In the 2nd paragraph, Page 2, and the 2nd paragraph, Page 3, the authors list several studies by only mentioning their work or results. A more in-depth introduction is expected to present readers with some useful information.

4.       Although several membrane processes have been used for water purifications, their fouling issues are quite different from each other, depending on the feed source, kind and concentration of impurity, operation conditions, and membrane types. Therefore, experiences from other membrane processes, as the authors listed on Page 3, may not be suitable for MD.

5.       In line 107, “the particles of dispersed phase in the gel connect with each other to form a structure in the whole system,…”. What do the particles here refer to? If they form the backbone of gel, they are no longer dispersed phase, right?

6.       What is exactly the composition of the employed gel? From Table 1, the ion concentrations of various ions in two seawater are quite different from each other. Also, why do the gels need to be placed with an anhydrous CaCl2 solution, during preparation?

7.       Why authors chose alginic acid to humic acid ratio of 1:1? Also, what is the point to show readers Fig. 1A and Fig. 1B?

8.       In MD, the feed, where gel pollutant is formed, is preheated to a desirable temperature, so the gel should have the same temperature as the feed, or at least the majority part contacting feed. The heat transport among gels should be unidirectional along the cross-section (i.e., from feed side to membrane surface). However, in the current work, the temperature of gels was set to 301K, which is lower than the operating temperature of MD, and the heat transport was from surfaces to the center.

9.        As discussed in 4.1, XRD only provided information about the type of salt, which should be known before running XRD, since the gel was prepared by the authors.

10.   In Table 2, what does “Different concentrations of organic acid samples” mean?

11.   Line 445, “The weight loss rate in the third and fourth stages was slow, indicating the sufficient thermal stability of the gels”. The reason that the weight loss rate in these two stages was slow is highly possible because non-water compound only counts for less than 40%, and therefore the weight loss rate of them is less significant based on original gel mass.  

12.   In real conditions, gels should be flushed from the membrane surface by an external force, which is quite different from that in mechanic property tests, where the sample strip is pulled in two opposite directions.  However, the authors concluded gels existed good flushing resistance simply from stress results.

13.   In Line 485, “However, in the actual system operation, adding inhibitors will change the osmotic pressure of seawater, affecting the conventional operation parameters, and even necessitate an increased pressure in the equipment for processing, which is not worth the loss.” To the reader’s knowledge, osmotic pressure should not affect the operation in MD.

14.   The caption of Fig. 6A is mismatched with the title of the y-axis. How authors calculated the density/dense degree of gels? It is not stated in the experiment section.

15.   Line 524, the thermal conductivity of gel ranged from 0.028 to 1.154 Wm-1K-1, which is a very large range.

16.   In line 584, “according to the “like dissolves like” rule, the gel and cleaning agent can be mutually soluble; that is, the gel is easily disintegrable”. Soluble does not suggest disintegrable.

17.   Line 603, how do gels and salt (cleaning agent) penetrate each other?

18.   Authors should pay some attention to the writing since there existed plenty of grammar errors or confusing sentences. A few are listed here:

a.       Line 81: “Membrane cleaning is mainly to destroy the solute layer adsorbed on the membrane surface or remove impurities in the membrane channel, thereby restoring the original membrane flux as far as possible”.

b.       Line 266: “First, four gel polymer chains which with 96 repeated units for each were randomly 266 packed in an initial simulation box (the cell) of 5.1 × 5.1 × 4.5 nm3 in size.”

c.       Line 113: “Based on this, the action mechanism of the cleaning agent on the gel fouling layer is not clear”

Author Response

We have studied the comments from you carefully and have incorporated the suggested changes into the manuscript to the best of my ability. We have made extensive modifications to our manuscript and supplemented extra data to make our results convincing. The point to respond to your comments are listed as following. To facilitate your review of our revisions, the following is a point-by-point response to the questions and comments delivered in your letter.

Reviewer 3:

Question 1:

What are the differences between salt washing and chemical cleaning?  The most chemical cleaning agents used in RO or NF are NaOH or acid, which are also considered salt.

Response:

Chemical cleaning agents mentioned in the literature typically include alkaline solutions (e.g., NaOH), metal chelating agents (e.g., EDTA), and surfactants (e.g., SDS) [1]. However, chemical cleaning processes can reduce the lifetime of membrane by damaging its active layer; they can also negatively impact the environment when chemical reagents are discharged as wastewater. Therefore, eco-friendly cleaning techniques have been developed to reduce or even eliminate the need for chemical cleaning. Salt cleaning has recently been suggested as alternatives to chemical cleaning, owing to their environmental friendliness, low cost application, and high cleaning efficiency.

Question 2:

Membrane distillation is commonly used for saline feed, which is too harsh for RO, with potential higher pollutant concentration. However, in the current work, the authors used simulated “seawater” which is the feed for RO, not MD, as feed. Can the authors explain the reason behind it?

Response:

The background of this study is the membrane-based humidification/dehumidification seawater desalination (MHDD) system (a type of membrane distillation), which has emerged as a new water treatment technology that combines thermal and membrane methods. MHDD, unlike RO, can separate feed solutions at temperatures well below the boiling point and at atmospheric pressure. The typical supply temperature is around 40-70°C, the residual heat flow can be reused, and alternative energy sources such as sun, wind, and geothermal can be used. In addition, MD is less susceptible to flux limitation due to concentration polarization than RO and can theoretically provide 100% retention for nonvolatile dissolved substances. The reason why MHDD often uses artificial seawater is that the seawater has complex composition and unstable characteristics. Artificial seawater is stable and repeatable, and can be adjusted according to different composition.

Question 3:

In the 2nd paragraph, Page 2, and the 2nd paragraph, Page 3, the authors list several studies by only mentioning their work or results. A more in-depth introduction is expected to present readers with some useful information.

Response:

We are grateful for the suggestions,We have made some additions to the content.

In the 2nd paragraph, Page 2, we have added the following information: Choudhury et al. [7] examined the interactions between contaminants and different types of MD membranes and the influence of different operating conditions on the occurrence of fouling and wetting. Membrane wetting will let the direct permeation of the feed solution through the membrane pores, results in reduced contaminant rejection and overall process failure. Jia et al. [8] attached a thin super-hydrophilic coating to the superhydrophobic film to construct a film with an asymmetric structure, which reduced pollution and scaling, and exhibited higher and more stable vapor flux than the original PTFE film. Qin et al. [9] examined the synergistic effect of combined fouling in MD process with three organic foulants in the presence of colloidal silica particles. Results showed that the combined organic fouling with colloidal silica particle not only deteriorated water production, but also compromised product quality by partial membrane wetting. Shao et al. [11] carried out appropriate charge modification of neutral ultrafiltration membrane to better reduce membrane pollution caused by electrostatic interaction and the comprehensive effect of membrane aperture.

In the 2nd paragraph, Page 3, we have added the following information: Stewart et al. [26] studied the binding of sodium and calcium ions to single and multiple poly-G decamer strands by conducting a series of molecular dynamics simulations. The results revealed the binding modes that provide a rationale for the observed gelling of alginate by calcium rather than sodium ions. Hecht et al. [27] used molecular dynamics simulation to investigate the binding of sodium and calcium ions to alginate chains, as well as its dependence on the content of guluronic (G) acid residues. The results revealed the association behavior that has a clear dependence on G content and the nature of interactions of sodium and calcium ions are shown to differ for poly-M, poly-G, and the heteropolymer compositions. Stewart et al. [29] studied the effect of the alginate amphiphilic structural on membrane fouling by molecular dynamics. Quantum chemical calculations on the M and G monomers of alginate reveal that M adopts an equilibrium geometry that is hydrophilic on one face and hydrophobic on the other, which is potentially amphiphilic.

Question 4:

Although several membrane processes have been used for water purifications, their fouling issues are quite different from each other, depending on the feed source, kind and concentration of impurity, operation conditions, and membrane types. Therefore, experiences from other membrane processes, as the authors listed on Page 3, may not be suitable for MD.

Response:

We have deleted references [16] and [22] on Page 3 that refer to other membrane processes as suggested by you.

Question 5:

In line 107, “the particles of dispersed phase in the gel connect with each other to form a structure in the whole system,…”. What do the particles here refer to? If they form the backbone of gel, they are no longer dispersed phase, right?

Response:

We are grateful for the suggestions. We have modified the expression as you suggested as follows: The macromolecular organic acid particles in the dispersed phase in the solution are interconnected by divalent metal ions such as Ca2+ to form a three dimensional network structure in the whole system, and the liquid is wrapped in it.

Question 6:

What is exactly the composition of the employed gel? From Table 1, the ion concentrations of various ions in two seawater are quite different from each other. Also, why do the gels need to be placed with an anhydrous CaCl2 solution, during preparation?

Response:

The skeleton of the gel was calcium alginate and calcium humate, and the internal solution was a mixed solution. The composition of gels from different seawater varies, but the preparation method is the same. The chemical crosslinking method mentioned in this study can prepare gels according to different composition and ratio. The gel was placed together with anhydrous CaCl2 solution because CaCl2, as the cross-linking agent of the gel itself, had minimal impact on the gel, which can prevent the gel from losing water and other situations caused by the influence of external environment, maintained the stable state of the gel, and better carry out subsequent testing and characterization.

Question 7:

Why authors chose alginic acid to humic acid ratio of 1:1? Also, what is the point to show readers Fig. 1A and Fig. 1B?

Response:

There are differences in the ratio of alginate to humic acid in different seawater, and the choice of 1:1 is to comprehensively consider the properties of alginate and humic acid and eliminate the uncertainty caused by different dosage.

We have deleted the irrelevant images (Fig. 1A and Fig. 1B).

Question 8:

In MD, the feed, where gel pollutant is formed, is preheated to a desirable temperature, so the gel should have the same temperature as the feed, or at least the majority part contacting feed. The heat transport among gels should be unidirectional along the cross-section (i.e., from feed side to membrane surface). However, in the current work, the temperature of gels was set to 301K, which is lower than the operating temperature of MD, and the heat transport was from surfaces to the center.8.

Response:

We have re-set the gel temperature of the MD process to the same temperature as the feed solution (333.15K). To eliminate the instability of the initial structure, it was geometrically optimized and annealed (from 333.15K to 500K with a period of 1000ps). After many times of optimization, the calculation reaches convergence, and the calculated property parameters (density and thermal conductivity) of the gel are close to the experimental values.

Question 9:

As discussed in 4.1, XRD only provided information about the type of salt, which should be known before running XRD, since the gel was prepared by the authors.

Response:

In addition to comparison with the references, the analysis of the characteristic peaks of XRD results also show that: It can be seen that there was no obvious overlap or annexation of each peak zone.  This indicates that there was no significant chemical reaction between the substances free from the gel skeleton and the frame substance, and that they maintain their own properties, which should have an impact on the physical, chemical and permeability properties of the gel.  According to the unique water absorption and retention properties of the gel, the more substances trapped inside the gel, the higher the osmotic pressure, the more likely the gel will swell and increase the thickness. The thickening of the gel fouling layer is obviously detrimental to the moisture permeability of the permeable membrane.

Question 10:

In Table 2, what does “Different concentrations of organic acid samples” mean?.

Response:

Here, it means gels contain different concentrations of organic acids. We have revised it to a more appropriate description: Different samples (a-e).

Question 11:

Line 445, “The weight loss rate in the third and fourth stages was slow, indicating the sufficient thermal stability of the gels”. The reason that the weight loss rate in these two stages was slow is highly possible because non-water compound only counts for less than 40%, and therefore the weight loss rate of them is less significant based on original gel mass.

Response:

According to the calculation, the weight loss rate of the gel was 11% from 200 – 900 °C, which only accounts for 24% of the non-water compounds (which accounts for 45% of the gel), indicating that the gel has sufficient thermal stability.

Question 12:

In real conditions, gels should be flushed from the membrane surface by an external force, which is quite different from that in mechanic property tests, where the sample strip is pulled in two opposite directions.  However, the authors concluded gels existed good flushing resistance simply from stress results.12.

Response:

We have modified the description of the mechanical properties of the gel according to your suggestion: It also shows that the gel layer has good mechanical properties which can maintain its shape to a certain extent and the gel layer is not easily deformed by external influences.

Question 13:

In Line 485, “However, in the actual system operation, adding inhibitors will change the osmotic pressure of seawater, affecting the conventional operation parameters, and even necessitate an increased pressure in the equipment for processing, which is not worth the loss.” To the reader’s knowledge, osmotic pressure should not affect the operation in MD.

Response:

We have removed the inappropriate statements: affecting the conventional operation parameters. And we have made following modifications to this sentence according to your suggestions: However, in the actual system operation, adding inhibitors will change the osmotic pressure of seawater, and the pressure in the equipment for processing needs to be increased, which is not worth the loss.

Question 14:

The caption of Fig. 6A is mismatched with the title of the y-axis. How authors calculated the density/dense degree of gels? It is not stated in the experiment section.

Response:

We have added relevant content in the experimental section: Dense degree is defined as the mass of the substance formed after the inhibitor was added divided by the mass of the gel. The disintegration degree is defined as the dissolved mass divided by the total mass of gel.

Question 15:

Line 524, the thermal conductivity of gel ranged from 0.028 to 1.154 Wm-1K-1, which is a very large range.

Response:

We have done a lot of characterization, and the experimental value of the thermal conductivity of the gel is scattered, there is no law to follow. This may be related to the soft material properties of gels: Unlike solid heat conduction, which is mainly caused by the excitation and transmission of phonons, in soft matter, heat propagation is caused and propagated by the vibration, twisting and rotation of molecular chains. Such molecular-level interactions include intramolecular and intermolecular forces, which come from two sources: one is vibration or rotation of bond lengths and bond angles; The other is the interaction of non-bonded parts of the system, which mainly reflects the contribution of these energy forms through the change of bonds and the motion of atoms [2]. Which needs further study in the future.

Question 16:

In line 584, “according to the “like dissolves like” rule, the gel and cleaning agent can be mutually soluble; that is, the gel is easily disintegrable”. Soluble does not suggest disintegrable.

Response:

We have deleted “that is, the gel is easily disintegrable” according to your suggestion.

Question 17:

Line 603, how do gels and salt (cleaning agent) penetrate each other?

Response:

Here we refer to the process of dissolution, which we have modified to describe more appropriately: the gel and cleaning agent gradually came into contact and the tight structure of the gel stretches.

Question 18:

Authors should pay some attention to the writing since there existed plenty of grammar errors or confusing sentences. A few are listed here:

  1. Line 81: “Membrane cleaning is mainly to destroy the solute layer adsorbed on the membrane surface or remove impurities in the membrane channel, thereby restoring the original membrane flux as far as possible”.
  2. Line 266: “First, four gel polymer chains which with 96 repeated units for each were randomly packed in an initial simulation box (the cell) of 5.1 × 5.1 × 4.5 nm3 in size.”
  3. Line 113: “Based on this, the action mechanism of the cleaning agent on the gel fouling layer is not clear”

Response:

We are grateful for the suggestions. We have made modifications according to your suggestions:

  1. Membrane cleaning is mainly to restore the initial flux of the membrane as much as possible by destroying the solute layer adsorbed on the membrane surface or removing impurities in the membrane pores.
  2. First, molecules mentioned above were randomly packed in an initial simulation unit with a size of1 × 5.1 × 4.5 nm3.
  3. Based on the above reasons, the action mechanism of the cleaning agent on the gel fouling layer is still unclear.

Reviewer 4 Report

The research paper has a very good scientific level. The content of the paper is very valuable from the theoretical and practical point of view. The article presents a current topic. The aim of research is interesting and beneficial for this research field. The topic is convenient for the scope of the journal. The title of scientific article is clear and it sufficiently reflects content. The abstract and key words are informative. The figures appropriately complement the presentation of the scientific work results. The tables are acceptable in this state. The manuscript is well organized however there are some irregularities which need corrections. Some questions that arise from the text should be answered.

 1.  - Introduction: The aim of the research is obvious from the context, but I recommend emphasizing it.

2.       I recommend adding units of physical quantities to the description of equations in the text.

3.       Chapter 3.2: Please, explain more precisely how the temperature range was chosen for the thermal analysis.

4.       Why was Fourier's equation (11) used in one-dimensional form?

5.       The abbreviation of physical unit “liter” should be written in form “l” (not with capital letter “L”).

6.       I recommend using en dash instead of a hyphen to write a range of values (0 – 100) °C, etc. The physical unit should be written behind the brackets.

7.       The keywords should not be the same as those used in the title of the article. Keywords should start with a lowercase letter.

8.       Lines: 166, 435 – 437, 440, 442, 444 etc. The gap between the numerical value of the quantity and the unit should be added.

9.       The previously mentioned formal incorrections should be checked and corrected throughout the text.      

Author Response

We have studied the comments from you carefully and have incorporated the suggested changes into the manuscript to the best of my ability. We have made extensive modifications to our manuscript and supplemented extra data to make our results convincing. The point to respond to your comments are listed as following. To facilitate your review of our revisions, the following is a point-by-point response to the questions and comments delivered in your letter.

Reviewer 4:

Question 1:

Introduction: The aim of the research is obvious from the context, but I recommend emphasizing it.

Response:

We are appreciate your advice. We have added relevant literatures in the introduction to draw out the importance of regulating gel contaminants: Choudhury et al. [7] examined the interactions between contaminants and different types of MD membranes and the influence of different operating conditions on the occurrence of fouling and wetting. Membrane wetting will let the direct permeation of the feed solution through the membrane pores, results in reduced contaminant rejection and overall process failure. Hecht et al. [27] used molecular dynamics simulation to investigate the binding of sodium and calcium ions to alginate chains, as well as its dependence on the content of guluronic (G) acid residues. The results revealed the association behavior that has a clear dependence on G content and the nature of interactions of sodium and calcium ions are shown to differ for poly-M, poly-G, and the heteropolymer compositions. Stewart et al. [29] studied the effect of the alginate amphiphilic structural on membrane fouling by molecular dynamics. Quantum chemical calculations on the M and G monomers of alginate reveal that M adopts an equilibrium geometry that is hydrophilic on one face and hydrophobic on the other, which is potentially amphiphilic.

Question 2:

I recommend adding units of physical quantities to the description of equations in the text.

Response:

We have added the missing units of the physical quantity: E (KJ/mol); δ ((KJ/cm3)^1/2); CED (KJ/cm3), ΔE (KJ);  (cm3); ΔH (KJ/mol); ΔG (KJ/mol); ΔS (KJ/mol·K); Jz (W/m2); t (s); A (m2); m (g/mol); λ (W/m·K).

Question 3:

Chapter 3.2: Please, explain more precisely how the temperature range was chosen for the thermal analysis.

Response:

The thermogravimetric results of the gel are shown in Figure 4. According to the four fluctuating trends of DSC curve, which were obvious descending - ascending - steady ascending - slow descending, the cleavage of the gel could be divided into four stages.

Figure 4. The thermal reactivity of the gel layer.

Question 4:

Why was Fourier's equation (11) used in one-dimensional form?

Response:

Heat transfer is carried to the gel and membrane by a heat source on the hot seawater side, and the heat is transferred in the direction perpendicular to the membrane. To simplify the calculation, the one-dimensional form of the Fourier equation was used here.

Question 5:

The abbreviation of physical unit “liter” should be written in form “l” (not with capital letter “L”).

Response:

We have changed the "L" into "l" as you suggested. See line 152 and line 181of the main text for details.

Question 6:

I recommend using en dash instead of a hyphen to write a range of values (0 – 100) °C, etc. The physical unit should be written behind the brackets.

Response:

We have followed your suggestion to write a range of values using a dash instead of a hyphen, with the physical unit following the brackets. See line 173, line 419, line 456, line 474, line 477, line 479, line 482, and line 509 of the main text for details.

Question 7:

The keywords should not be the same as those used in the title of the article. Keywords should start with a lowercase letter.

Response:

We have changed the first letter of the keywords to lowercase as you suggested: seawater desalination; artificial contaminant gel; molecular dynamics simulation; effects of detergent.

Question 8:

Lines: 166, 435 – 437, 440, 442, 444 etc. The gap between the numerical value of the quantity and the unit should be added.

Response:

We have added gap between values and units as you suggested.

Question 9:

The previously mentioned formal incorrections should be checked and corrected throughout the text.

Response:

Thanks for your guidance, we have reviewed the entire text and corrected the errors mentioned.

Round 2

Reviewer 3 Report

After being carefully revised, the current version is suggested to be published in Polymers